# Volume Rendering of Neural Implicit Surfaces

**Lior Yariv**[1]      **Jiatao Gu**[2]      **Yoni Kasten**[1]      **Yaron Lipman**[1,2]

[1]Weizmann Institute of Science      [2]Facebook AI Research

## Abstract

Neural volume rendering became increasingly popular recently due to its success in synthesizing novel views of a scene from a sparse set of input images. So far, the geometry learned by neural volume rendering techniques was modeled using a generic density function. Furthermore, the geometry itself was extracted using an arbitrary level set of the density function leading to a noisy, often low fidelity reconstruction. The goal of this paper is to improve geometry representation and reconstruction in neural volume rendering. We achieve that by modeling the volume density as a function of the geometry. This is in contrast to previous work modeling the geometry as a function of the volume density. In more detail, we define the volume density function as Laplace's cumulative distribution function (CDF) applied to a signed distance function (SDF) representation. This simple density representation has three benefits: (i) it provides a useful inductive bias to the geometry learned in the neural volume rendering process; (ii) it facilitates a bound on the opacity approximation error, leading to an accurate sampling of the viewing ray. Accurate sampling is important to provide a precise coupling of geometry and radiance; and (iii) it allows efficient unsupervised disentanglement of shape and appearance in volume rendering. Applying this new density representation to challenging scene multiview datasets produced high quality geometry reconstructions, outperforming relevant baselines. Furthermore, switching shape and appearance between scenes is possible due to the disentanglement of the two.

## 1 Introduction

Volume rendering [18] is a set of techniques that renders volume *density* in *radiance fields* by the so called volume rendering integral. It has recently been shown that representing both the density and radiance fields as neural networks can lead to excellent prediction of novel views by learning only from a sparse set of input images. This neural volume rendering approach, presented in [21] and developed by its follow-ups [34, 2] approximates the integral as alpha-composition in a differentiable way, allowing to learn simultaneously both from input images. Although this coupling indeed leads to good generalization of novel viewing directions, the density part is not as successful in faithfully predicting the scene's actual geometry, often producing noisy, low fidelity geometry approximation.

We propose VolSDF to devise a different model for the density in neural volume rendering, leading to better approximation of the scene's geometry while maintaining the quality of view synthesis. The key idea is to represent the density as a function of the signed distance to the scene's surface, see Figure 1. Such density function enjoys several benefits. First, it guarantees the existence of a well-defined surface that generates the density. This provides a useful inductive bias for disentangling density and radiance fields, which in turn provides a more accurate geometry approximation. Second, we show this density formulation allows bounding the approximation error of the opacity along rays. This bound is used to sample the viewing ray so to provide a faithful coupling of density and radiance field in the volume rendering integral. E.g., without such a bound the computed radiance along a ray (pixel color) can potentially miss or extend surface parts leading to incorrect radiance approximation.

35th Conference on Neural Information Processing Systems (NeurIPS 2021).

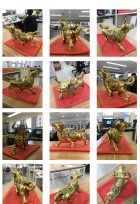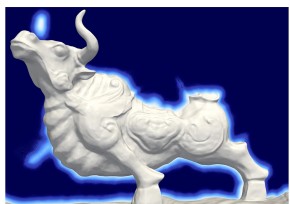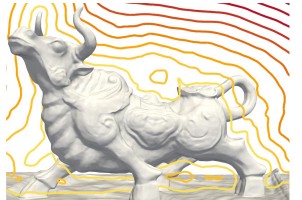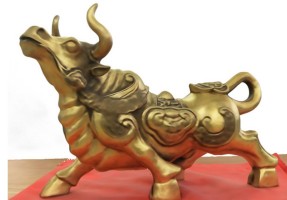

Figure 1: VolSDF: given a set of input images (left) we learn a volumetric density (center-left, sliced) defined by a signed distance function (center-right, sliced) to produce a neural rendering (right). This definition of density facilitates high quality geometry reconstruction (gray surfaces, middle).

A closely related line of research, often referred to as neural implicit surfaces [22, 38, 14], have been focusing on representing the scene's geometry implicitly using a neural network, making the surface rendering process differentiable. The main drawback of these methods is their requirement of masks that separate objects from the background. Also, learning to render surfaces directly tends to grow extraneous parts due to optimization problems, which are avoided by volume rendering. In a sense, our work combines the best of both worlds: *volume rendering with neural implicit surfaces*.

We demonstrate the efficacy of VolSDF by reconstructing surfaces from the DTU [12] and Blended-MVS [37] datasets. VolSDF produces more accurate surface reconstructions compared to NeRF [21] and NeRF++ [39], and comparable reconstruction compared to IDR [38], while avoiding the use of object masks. Furthermore, we show disentanglement results with our method, i.e., switching the density and radiance fields of different scenes, which is shown to fail in NeRF-based models.

## 2 Related work

**Neural Scene Representation & Rendering** Implicit functions are traditionally adopted in modeling 3D scenes [24, 11, 4]. Recent studies have been focusing on model implicit functions with multi-layer perceptron (MLP) due to its expressive representation power and low memory foot-print, including scene (geometry & appearance) representation [9, 20, 19, 23, 25, 29, 36, 28, 35] and free-view rendering [33, 16, 30, 26, 17, 21, 15, 39, 34, 2]. In particular, NeRF [21] has opened up a line of research (see [6] for an overview) combining neural implicit functions together with volume rendering to achieve photo-realistic rendering results. However, it is non-trivial to find a proper threshold to extract surfaces from the predicted density, and the recovered geometry is far from satisfactory. Furthermore, sampling of points along a ray for rendering a pixel is done using an opacity function that is approximated from another network without any guarantee for correct approximation.

**Multi-view 3D Reconstruction** Image-based 3D surface reconstruction (multi-view stereo) has been a longstanding problem in the past decades. Classical multi-view stereo approaches are generally either depth-based [1, 31, 8, 7] or voxel-based [5, 3, 32]. For instance, in COLMAP [31] (a typical depth-based method) image features are extracted and matched across different views to estimate depth. Then the predicted depth maps are fused to obtain dense point clouds. To obtain the surface, an additional meshing step e.g. Poisson surface reconstruction [13] is applied. However, these methods with complex pipelines may accumulate errors at each stage and usually result in incomplete 3D models, especially for non-Lambertian surfaces as they can not handle view dependent colors. On the contrary, although it produces complete models by directly modeling objects in a volume, voxel-based approaches are limited to low resolution due to high memory consumption. Recently, neural-based approaches such as DVR [22], IDR [38], NLR [14] have also been proposed to reconstruct scene geometry from multi-view images. However, these methods require accurate object masks and appropriate weight initialization due to the difficulty of propagating gradients.

Independently from and concurrently with our work here, [27] also use implicit surface representation incorporated into volume rendering. In particular, they replace the local transparency function with an occupancy network [19]. This allows adding surface smoothing term to the loss, improving the quality of the resulting surfaces. Differently from their approach, we use signed distance representation, regularized with an Eikonal loss [38, 10] without any explicit smoothing term. Furthermore, we show that the choice of using signed distance allows bounding the opacity approximation error, facilitating the approximation of the volume rendering integral for the suggested family of densities.

# 3 Method

In this section we introduce a novel parameterization for volume density, defined as transformed signed distance function. Then we show how this definition facilitates the volume rendering process. In particular, we derive a bound of the error in the opacity approximation and consequently devise a sampling procedure for approximating the volume rendering integral.

## 3.1 Density as transformed SDF

Let the set $\Omega \subset \mathbb{R}^3$ represent the space occupied by some object in $\mathbb{R}^3$, and $\mathcal{M} = \partial\Omega$ its boundary surface. We denote by $\mathbf{1}_\Omega$ the $\Omega$ indicator function, and by $d_\Omega$ the Signed Distance Function (SDF) to its boundary $\mathcal{M}$,

$$\mathbf{1}_\Omega(\boldsymbol{x}) = \begin{cases} 1 & \text{if } \boldsymbol{x} \in \Omega \\ 0 & \text{if } \boldsymbol{x} \notin \Omega \end{cases}, \quad \text{and } d_\Omega(\boldsymbol{x}) = (-1)^{\mathbf{1}_\Omega(\boldsymbol{x})} \min_{\boldsymbol{y} \in \mathcal{M}} \|\boldsymbol{x} - \boldsymbol{y}\|, \tag{1}$$

where $\|\cdot\|$ is the standard Euclidean 2-norm. In neural volume rendering the volume *density* $\sigma$ : $\mathbb{R}^3 \to \mathbb{R}_+$ is a scalar volumetric function, where $\sigma(\boldsymbol{x})$ is the rate that light is occluded at point $\boldsymbol{x}$; $\sigma$ is called density since it is proportional to the particle count per unit volume at $\boldsymbol{x}$ [18]. In previous neural volumetric rendering approaches [21, 15, 39], the density function, $\sigma$, was modeled with a general-purpose Multi-Layer Perceptron (MLP). In this work we suggest to model the density using a certain transformation of a learnable Signed Distance Function (SDF) $d_\Omega$, namely

$$\sigma(\boldsymbol{x}) = \alpha \Psi_\beta \left( -d_\Omega(\boldsymbol{x}) \right), \tag{2}$$

where $\alpha, \beta > 0$ are learnable parameters, and $\Psi_\beta$ is the Cumulative Distribution Function (CDF) of the Laplace distribution with zero mean and $\beta$ scale (i.e., mean absolute deviation, which is intuitively the $L_1$ version of the standard deviation),

$$\Psi_\beta(s) = \begin{cases} \frac{1}{2} \exp\left(\frac{s}{\beta}\right) & \text{if } s \leq 0 \\ 1 - \frac{1}{2} \exp\left(-\frac{s}{\beta}\right) & \text{if } s > 0 \end{cases} \tag{3}$$

Figure 1 (center left and right) depicts an example of such a density and SDF. As can be readily checked from this definition, as $\beta$ approach zero, the density $\sigma$ converges to a scaled indicator function of $\Omega$, that is $\sigma \to \alpha \mathbf{1}_\Omega$ for all points $\boldsymbol{x} \in \Omega \setminus \mathcal{M}$.

Intuitively, the density $\sigma$ models a homogeneous object with a constant density $\alpha$ that smoothly decreases near the object's boundary, where the smoothing amount is controlled by $\beta$. The benefit in defining the density as in equation 2 is two-fold: First, it provides a useful inductive bias for the surface geometry $\mathcal{M}$, and provides a principled way to reconstruct the surface, i.e., as the zero level-set of $d_\Omega$. This is in contrast to previous work where the reconstruction was chosen as an arbitrary level set of the learned density. Second, the particular form of the density as defined in equation 2 facilitates a bound on the error of the *opacity* (or, equivalently the *transparency*) of the rendered volume, a crucial component in the volumetric rendering pipeline. In contrast, such a bound will be hard to devise for a generic MLP densities.

## 3.2 Volume rendering of $\sigma$

In this section we review the volume rendering integral and the numerical integration commonly used to approximate it, requiring a set $\mathcal{S}$ of sample points per ray. In the following section (Section 3.3), we explore the properties of the density $\sigma$ and derive a bound on the opacity approximation error along viewing rays. Finally, in Section 3.4 we derive an algorithm for producing a sample $\mathcal{S}$ to be used in the volume rendering numerical integration.

In volume rendering we consider a ray $\boldsymbol{x}$ emanating from a camera position $\boldsymbol{c} \in \mathbb{R}^3$ in direction $\boldsymbol{v} \in \mathbb{R}^3$, $\|\boldsymbol{v}\| = 1$, defined by $\boldsymbol{x}(t) = \boldsymbol{c} + t\boldsymbol{v}$, $t \geq 0$. In essence, volume rendering is all about approximating the integrated (i.e., summed) light radiance along this ray reaching the camera. There are two important quantities that participate in this computation: the volume's *opacity* $O$, or equivalently, its *transparency* $T$, and the *radiance field* $L$.

The *transparency* function of the volume along a ray $\boldsymbol{x}$, denoted $T$, indicates, for each $t \geq 0$, the probability a light particle succeeds traversing the segment $[\boldsymbol{c}, \boldsymbol{x}(t)]$ without bouncing off,

$$T(t) = \exp\left(-\int_0^t \sigma(\boldsymbol{x}(s))ds\right), \tag{4}$$

and the *opacity* $O$ is the complement probability,

$$O(t) = 1 - T(t). \tag{5}$$

Note that $O$ is a monotonic increasing function where $O(0) = 0$, and assuming that every ray is eventually occluded $O(\infty) = 1$. In that sense we can think of $O$ as a CDF, and

$$\tau(t) = \frac{dO}{dt}(t) = \sigma(\boldsymbol{x}(t))T(t) \tag{6}$$

is its Probability Density Function (PDF). The volume rendering equation is the expected light along the ray,

$$I(\boldsymbol{c}, \boldsymbol{v}) = \int_0^\infty L(\boldsymbol{x}(t), \boldsymbol{n}(t), \boldsymbol{v})\tau(t)dt, \tag{7}$$

where $L(\boldsymbol{x}, \boldsymbol{n}, \boldsymbol{v})$ is the radiance field, namely the amount of light emanating from point $\boldsymbol{x}$ in direction $\boldsymbol{v}$; in our formulation we also allow $L$ to depend on the level-set's normal, i.e., $\boldsymbol{n}(t) = \nabla_{\boldsymbol{x}}d_\Omega(\boldsymbol{x}(t))$. Adding this dependency is motivated by the fact that BRDFs of common materials are often encoded with respect to the surface normal, facilitating disentanglement as done in surface

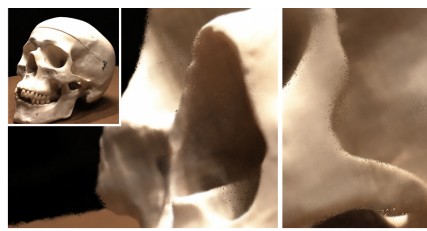

NeRF

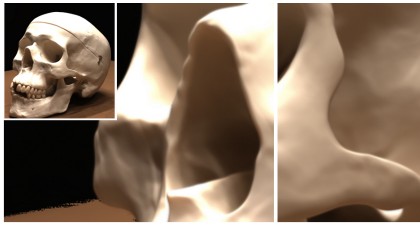

**VolSDF**

Figure 2: Qualitative comparison to NeRF. VolSDF shows less artifacts.

rendering [38]. We will get back to disentanglement in the experiments section. The integral in equation 7 is approximated using a numerical quadrature, namely the rectangle rule, at some discrete samples $\mathcal{S} = \{s_i\}_{i=1}^m$, $0 = s_1 < s_2 < \ldots < s_m = M$, where $M$ is some large constant:

$$I(\boldsymbol{c}, \boldsymbol{v}) \approx \hat{I}_\mathcal{S}(\boldsymbol{c}, \boldsymbol{v}) = \sum_{i=1}^{m-1} \hat{\tau}_i L_i, \tag{8}$$

where we use the subscript $\mathcal{S}$ in $\hat{I}_\mathcal{S}$ to highlight the dependence of the approximation on the sample set $\mathcal{S}$, $\hat{\tau}_i \approx \tau(s_i)\Delta s$ is the approximated PDF multiplied by the interval length, and $L_i = L(\boldsymbol{x}(s_i), \boldsymbol{n}(s_i), \boldsymbol{v})$ is the sampled radiance field. We provide full derivation and detail of $\hat{\tau}_i$ in the supplementary.

**Sampling.** Since the PDF $\tau$ is typically extremely concentrated near the object's boundary (see e.g., Figure 3, right) the choice of the sample points $\mathcal{S}$ has a crucial effect on the approximation quality of equation 8. One solution is to use an adaptive sample, e.g., $\mathcal{S}$ computed with the inverse CDF, i.e., $O^{-1}$. However, $O$ depends on the density model $\sigma$ and is not given explicitly. In [21] a second, coarse network was trained specifically for the approximation of the opacity $O$, and was used for inverse sampling. However, the second network's density does not necessarily faithfully represents the first network's density, for which we wish to compute the volume integral. Furthermore, as we show later, one level of sampling could be insufficient to produce an accurate sample $\mathcal{S}$. Using a naive or crude approximation of $O$ would lead to a sub-optimal sample set $\mathcal{S}$ that misses, or over extends non-negligible $\tau$ values. Consequently, incorrect radiance approximations can occur (i.e., pixel color), potentially harming the learned density-radiance field decomposition. Our solution works with a single density $\sigma$, and the sampling $\mathcal{S}$ is computed by a sampling algorithm based on an error bound for the opacity approximation. Figure 2 compares the NeRF and VolSDF renderings for the same scene. Note the salt and pepper artifacts in the NeRF rendering caused by the random samples; using fixed (uniformly spaced) sampling in NeRF leads to a different type of artifacts shown in the supplementary.

### 3.3 Bound on the opacity approximation error

In this section we develop a bound on the opacity approximation error using the rectangle rule. For a set of samples $\mathcal{T} = \{t_i\}_{i=1}^n$, $0 = t_1 < t_2 < \cdots < t_n = M$, we let $\delta_i = t_{i+1} - t_i$, and $\sigma_i = \sigma(\boldsymbol{x}(t_i))$. Given some $t \in (0, M]$, assume $t \in [t_k, t_{k+1}]$, and apply the rectangle rule (i.e., left Riemann sum) to get the approximation:

$$\int_0^t \sigma(\boldsymbol{x}(s))ds = \widehat{R}(t) + E(t), \quad \text{where} \ \ \widehat{R}(t) = \sum_{i=1}^{k-1} \delta_i \sigma_i + (t - t_k)\sigma_k \tag{9}$$

is the rectangle rule approximation, and $E(t)$ denotes the error in this approximation. The corresponding approximation of the opacity function (equation 5) is

$$\widehat{O}(t) = 1 - \exp\left(-\widehat{R}(t)\right). \tag{10}$$

Our goal in this section is to derive a uniform bound over $[0, M]$ to the approximation $\widehat{O} \approx O$. The key is the following bound on the derivative[1] of the density $\sigma$ inside an interval along the ray $\boldsymbol{x}(t)$:

**Theorem 1.** *The derivative of the density $\sigma$ within a segment $[t_i, t_{i+1}]$ satisfies*

$$\left|\frac{d}{ds}\sigma(\boldsymbol{x}(s))\right| \leq \frac{\alpha}{2\beta}\exp\left(-\frac{d_i^\star}{\beta}\right), \text{ where } d_i^\star = \min_{\substack{s \in [t_i, t_{i+1}] \\ \boldsymbol{y} \notin B_i \cup B_{i+1}}} \|\boldsymbol{x}(s) - \boldsymbol{y}\|, \tag{11}$$

*and $B_i = \{\boldsymbol{x} \mid \|\boldsymbol{x} - \boldsymbol{x}(t_i)\| < |d_i|\}$, $d_i = d_\Omega(\boldsymbol{x}(t_i))$.*

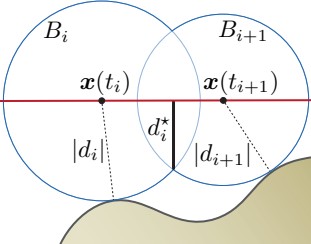

The proof of this theorem, which is provided in the supplementary, makes a principled use of the signed distance function's unique properties; the explicit formula for $d_i^*$ is a bit cumbersome and therefore is deferred to the supplementary as-well. The inset depicts the boundary of the open balls union $B_i \cup B_{i+1}$, the interval $[\boldsymbol{x}(t_i), \boldsymbol{x}(t_{i+1})]$ and the bound is defined in terms of the minimal distance between these two sets, i.e., $d_i^*$.

The benefit in Theorem 1 is that it allows to bound the density's derivative in each interval $[t_i, t_{i-1}]$ based only on the unsigned distance at the interval's end points, $|d_i|, |d_{i+1}|$, and the density parameters $\alpha, \beta$. This bound can be used to derive an error bound for the rectangle rule's approximation of the opacity,

$$|E(t)| \leq \widehat{E}(t) = \frac{\alpha}{4\beta}\left(\sum_{i=1}^{k-1}\delta_i^2 e^{-\frac{d_i^\star}{\beta}} + (t - t_k)^2 e^{-\frac{d_k^\star}{\beta}}\right). \tag{12}$$

Details are in the supplementary. Equation 12 leads to the following opacity error bound, also proved in the supplementary:

**Theorem 2.** *For $t \in [0, M]$, the error of the approximated opacity $\hat{O}$ can be bounded as follows:*

$$\left|O(t) - \widehat{O}(t)\right| \leq \exp\left(-\widehat{R}(t)\right)\left(\exp\left(\widehat{E}(t)\right) - 1\right) \tag{13}$$

Finally, we can bound the opacity error for $t \in [t_k, t_{k+1}]$ by noting that $\widehat{E}(t)$, and consequently also $\exp(\widehat{E}(t))$ are monotonically increasing in $t$, while $\exp(-\widehat{R}(t))$ is monotonically decreasing in $t$, and therefore

$$\max_{t \in [t_k, t_{k+1}]}\left|O(t) - \widehat{O}(t)\right| \leq \exp\left(-\widehat{R}(t_k)\right)\left(\exp(\widehat{E}(t_{k+1})) - 1\right). \tag{14}$$

Taking the maximum over all intervals furnishes a bound $B_{\mathcal{T}, \beta}$ as a function of $\mathcal{T}$ and $\beta$,

$$\max_{t \in [0, M]}\left|O(t) - \widehat{O}(t)\right| \leq B_{\mathcal{T}, \beta} = \max_{k \in [n-1]}\left\{\exp\left(-\widehat{R}(t_k)\right)\left(\exp(\widehat{E}(t_{k+1})) - 1\right)\right\}, \tag{15}$$

where by convention $\widehat{R}(t_0) = 0$, and $[\ell] = \{1, 2, \ldots, \ell\}$. See Figure 3, where this bound is visualized in faint-red.

To conclude this section we derive two useful properties, proved in the supplementary. The first, is that sufficiently dense sampling is guaranteed to reduce the error bound $B_{\mathcal{T}, \epsilon}$:

**Lemma 1.** *Fix $\beta > 0$. For any $\epsilon > 0$ a sufficient dense sampling $\mathcal{T}$ will provide $B_{\mathcal{T}, \beta} < \epsilon$.*

Second, with a fixed number of samples we can set $\beta$ such that the error bound is below $\epsilon$:

**Lemma 2.** *Fix $n > 0$. For any $\epsilon > 0$ a sufficiently large $\beta$ that satisfies*

$$\beta \geq \frac{\alpha M^2}{4(n-1)\log(1+\epsilon)} \tag{16}$$

*will provide $B_{\mathcal{T}, \beta} \leq \epsilon$.*

---

[1] As $d_\Omega$ is not differentiable everywhere the bound is on the *Lipschitz constant* of $\sigma$, see supplementary.

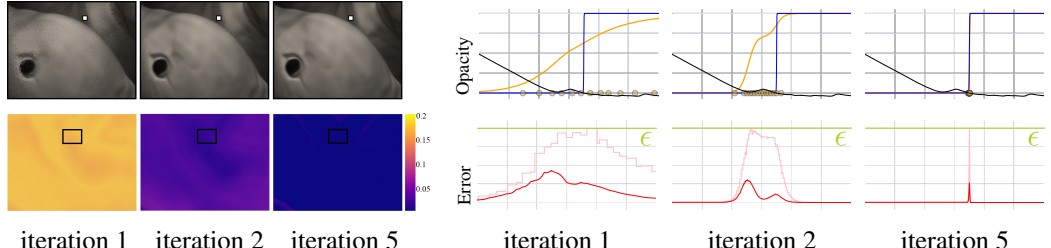

iteration 1   iteration 2   iteration 5      iteration 1      iteration 2      iteration 5

Figure 3: Qualitative evaluation of Algorithm 1 after 1, 2 and 5 iterations. Left-bottom: per-pixel $\beta_+$ heatmap; Left-top: rendering of areas marked with black squares. Right-top: for a single ray indicated by white pixel we show the approximated (orange), true opacity (blue), the SDF (black), and $\widehat{O}^{-1}$ sample example (yellow dots). Right-bottom: for the same ray we now show the true opacity error (red), and error bound (faint red). After 5 iterations most of the rays converged, as can be inspected by the blue colors in the heatmap, providing a guaranteed $\epsilon$ approximation to the opacity, resulting in a crisp and more accurate rendering (center-left, top).

## 3.4 Sampling algorithm

In this section we develop an algorithm for computing the sampling $\mathcal{S}$ to be used in equation 8. This is done by first utilizing the bound in equation 15 to find samples $\mathcal{T}$ so that $\widehat{O}$ (via equation 10) provides an $\epsilon$ approximation to the true opacity $O$, where $\epsilon$ is a hyper-parameter, that is $B_{\mathcal{T},\beta} < \epsilon$. Second, we perform inverse CDF sampling with $\hat{O}$, as described in Section 3.2.

Note that from Lemma 1 it follows that we can simply choose large enough $n$ to ensure $B_{\mathcal{T},\beta} < \epsilon$. However, this would lead to prohibitively large number of samples. Instead, we suggest a simple algorithm to reduce the number of required samples in practice and allows working with a limited budget of sample points. In a nutshell, we start with a uniform sampling $\mathcal{T} = \mathcal{T}_0$, and use Lemma 2 to initially set a $\beta_+ > \beta$ that satisfies $B_{\mathcal{T},\beta_+} \leq \epsilon$. Then, we repeatedly upsample $\mathcal{T}$ to reduce $\beta_+$ while maintaining $B_{\mathcal{T},\beta_+} \leq \epsilon$. Even though this simple strategy is not guaranteed to converge, we find that $\beta_+$ usually converges to $\beta$ (typically 85%, see also Figure 3), and even in cases it does not, the algorithm provides $\beta_+$ for which the opacity approximation still maintains an $\epsilon$ error. The algorithm is presented below (Algorithm 1).

We initialize $\mathcal{T}$ (Line 1 in Algorithm 1) with uniform sampling $\mathcal{T}_0 = \{t_i\}_{i=1}^n$, where $t_k = (k-1)\frac{M}{n-1}$, $k \in [n]$ (we use $n = 128$ in our implementation). Given this sampling we next pick $\beta_+ > \beta$ according to Lemma 2 so that the error bound satisfies the required $\epsilon$ bound (Line 2 in Algorithm 1).

In order to reduce $\beta_+$ while keep $B_{\mathcal{T},\beta_+} \leq \epsilon$, $n$ samples are added to $\mathcal{T}$ (Line 4 in Algorithm 1), where the number of points sampled from each interval is proportional to its current error bound, equation 14. Assuming $\mathcal{T}$ was sufficiently upsampled and satisfy $B_{\mathcal{T},\beta_+} < \epsilon$, we decrease $\beta_+$ towards $\beta$. Since the algorithm did not stop we have that $B_{\mathcal{T},\beta} > \epsilon$. Therefore the Mean Value Theorem implies the existence of $\beta_\star \in (\beta, \beta_+)$ such that $B_{\mathcal{T},\beta_\star} = \epsilon$. We use the bisection method (with maximum of 10 iterations) to efficiently search for $\beta_\star$ and update $\beta_+$ accordingly (Lines 6 and 7 in Algorithm 1). The algorithm runs iteratively until $B_{\mathcal{T},\beta} \leq \epsilon$ or a maximal number of 5 iterations is reached. Either way, we use the final $\mathcal{T}$ and $\beta_+$ (guaranteed to provide $B_{\mathcal{T},\beta_+} \leq \epsilon$) to estimate the current opacity $\widehat{O}$, Line 10 in Algorithm 1. Finally we return a fresh set of $m = 64$ samples $\hat{O}$ using inverse transform sampling (Line 11 in Algorithm 1). Figure 3 shows qualitative illustration of Algorithm 1, for $\beta = 0.001$ and $\epsilon = 0.1$ (typical values).

---

**Algorithm 1:** Sampling algorithm.

**Input:** error threshold $\epsilon > 0$; $\beta$

1  Initialize $\mathcal{T} = \mathcal{T}_0$

2  Initialize $\beta_+$ such that $B_{\mathcal{T},\beta_+} \leq \epsilon$

3  **while** $B_{\mathcal{T},\beta} > \epsilon$ *and not max_iter* **do**

4     upsample $\mathcal{T}$

5     **if** $B_{\mathcal{T},\beta_+} < \epsilon$ **then**

6        Find $\beta_\star \in (\beta, \beta_+)$ so that $B_{\mathcal{T},\beta_\star} = \epsilon$

7        Update $\beta_+ \leftarrow \beta_\star$

8     **end**

9  **end**

10  Estimate $\widehat{O}$ using $\mathcal{T}$ and $\beta_+$

11  $\mathcal{S} \leftarrow$ get fresh $m$ samples using $\hat{O}^{-1}$

12  **return** $\mathcal{S}$

---

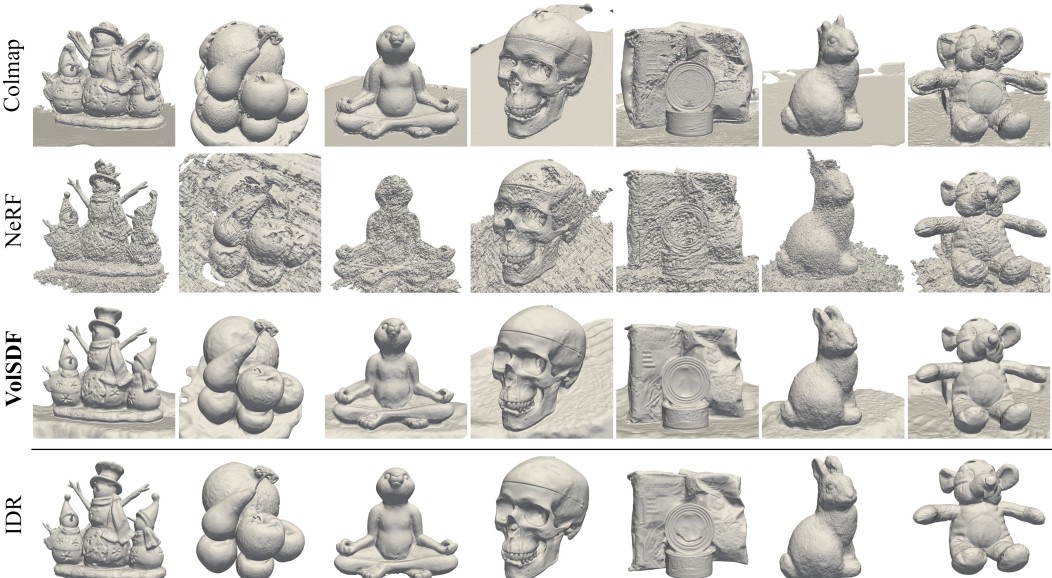

Figure 4: Qualitative results for reconstructed geometries of objects from the DTU dataset.

## 3.5 Training

Our system consists of two Multi-Layer Perceptrons (MLP): (i) $\boldsymbol{f}_\varphi$ approximating the SDF of the learned geometry, as well as global geometry feature $\boldsymbol{z}$ of dimension 256, i.e., $\boldsymbol{f}_\varphi(\boldsymbol{x}) = (d(\boldsymbol{x}), \boldsymbol{z}(\boldsymbol{x})) \in \mathbb{R}^{1+256}$, where $\varphi$ denotes its learnable parameters; (ii) $L_\psi(\boldsymbol{x}, \boldsymbol{n}, \boldsymbol{v}, \boldsymbol{z}) \in \mathbb{R}^3$ representing the scene's radiance field with learnable parameters $\psi$. In addition we have two scalar learnable parameters $\alpha, \beta \in \mathbb{R}$. In fact, in our implementation we make the choice $\alpha = \beta^{-1}$. We denote by $\theta \in \mathbb{R}^p$ the collection of all learnable parameters of the model, $\theta = (\varphi, \psi, \beta)$. To facilitate the learning of high frequency details of the geometry and radiance field, we exploit positional encoding [21] for the position $\boldsymbol{x}$ and view direction $\boldsymbol{v}$ in the geometry and radiance field. The influence of different positional encoding choices are presented in the supplementary.

Our data consists of a collection of images with camera parameters. From this data we extract pixel level data: for each pixel $p$ we have a triplet $(I_p, \boldsymbol{c}_p, \boldsymbol{v}_p)$, where $I_p \in \mathbb{R}^3$ is its intensity (RGB color), $\boldsymbol{c}_p \in \mathbb{R}^3$ is its camera location, and $\boldsymbol{v}_p \in \mathbb{R}^3$ is the viewing direction (camera to pixel). Our training loss consists of two terms:

$$\mathcal{L}(\theta) = \mathcal{L}_{\text{RGB}}(\theta) + \lambda \mathcal{L}_{\text{SDF}}(\varphi), \quad \text{where} \tag{17}$$

$$\mathcal{L}_{\text{RGB}}(\theta) = \mathbb{E}_p \left\| I_p - \hat{I}_{\mathcal{S}}(\boldsymbol{c}_p, \boldsymbol{v}_p) \right\|_1, \quad \text{and} \quad \mathcal{L}_{\text{SDF}}(\varphi) = \mathbb{E}_{\boldsymbol{z}} \left( \|\nabla d(\boldsymbol{z})\| - 1 \right)^2, \tag{18}$$

where $\mathcal{L}_{\text{RGB}}$ is the color loss; $\|\cdot\|_1$ denotes the 1-norm, $\mathcal{S}$ is computed with Algorithm 1, and $\hat{I}_{\mathcal{S}}$ is the numerical approximation to the volume rendering integral in equation 8; here we also incorporate the global feature in the radiance field, i.e., $L_i = L_\psi(\boldsymbol{x}(s_i), \boldsymbol{n}(s_i), \boldsymbol{v}_p, \boldsymbol{z}(\boldsymbol{x}(s_i)))$. $\mathcal{L}_{\text{SDF}}$ is the Eikonal loss encouraging $d$ to approximate a signed distance function [10]; the samples $\boldsymbol{z}$ are taken to combine a single random uniform space point and a single point from $\mathcal{S}$ for each pixel $p$. We train with batches of size 1024 pixels $p$. $\lambda$ is a hyper-parameter set to 0.1 throughout the the experiments. Further implementation details are provided in the supplementary.

## 4 Experiments

We evaluate our method on the challenging task of multiview 3D surface reconstruction. We use two datasets: DTU [12] and BlendedMVS [37], both containing real objects with different materials that are captured from multiple views. In Section 4.1 we show qualitative and quantitative 3D surface reconstruction results of VolSDF, comparing favorably to relevant baselines. In Section 4.2 we demonstrate that, in contrast to NeRF [21], our model is able to successfully disentangle the geometry and appearance of the captured objects.

| Scan | 24 | 37 | 40 | 55 | 63 | 65 | 69 | 83 | 97 | 105 | 106 | 110 | 114 | 118 | 122 | **Mean** |
|---|---|---|---|---|---|---|---|---|---|---|---|---|---|---|---|---|
| **Chamfer Distance** | | | | | | | | | | | | | | | | |
| **IDR** | 1.63 | 1.87 | 0.63 | 0.48 | 1.04 | 0.79 | 0.77 | 1.33 | 1.16 | 0.76 | 0.67 | 0.90 | 0.42 | 0.51 | 0.53 | 0.90 |
| colmap$_7$ | 0.45 | 0.91 | 0.37 | 0.37 | 0.90 | 1.00 | 0.54 | 1.22 | 1.08 | 0.64 | 0.48 | 0.59 | 0.32 | 0.45 | 0.43 | 0.65 |
| colmap$_0$ | **0.81** | 2.05 | **0.73** | 1.22 | 1.79 | 1.58 | 1.02 | 3.05 | 1.40 | 2.05 | 1.00 | 1.32 | 0.49 | 0.78 | 1.17 | 1.36 |
| **NeRF** | 1.92 | 1.73 | 1.92 | 0.80 | 3.41 | 1.39 | 1.51 | 5.44 | 2.04 | 1.10 | 1.01 | 2.88 | 0.91 | 1.00 | 0.79 | 1.89 |
| **VolSDF** | 1.14 | **1.26** | 0.81 | **0.49** | **1.25** | **0.70** | **0.72** | **1.29** | **1.18** | **0.70** | **0.66** | **1.08** | **0.42** | **0.61** | **0.55** | **0.86** |
| **PSNR** | | | | | | | | | | | | | | | | |
| **NeRF** | 26.24 | 25.74 | 26.79 | 27.57 | 31.96 | 31.50 | 29.58 | 32.78 | 28.35 | 32.08 | 33.49 | 31.54 | 31.0 | 35.59 | 35.51 | 30.65 |
| **VolSDF** | 26.28 | 25.61 | 26.55 | 26.76 | 31.57 | 31.5 | 29.38 | 33.23 | 28.03 | 32.13 | 33.16 | 31.49 | 30.33 | 34.9 | 34.75 | 30.38 |

Table 1: Quantitative results for the DTU dataset.

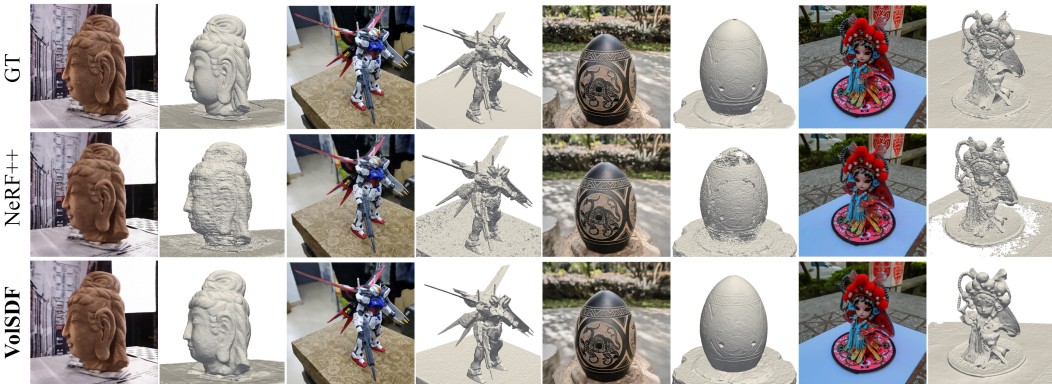

Figure 5: Qualitative results sampled from the BlendedMVS dataset. For each scan we present a visualization of a rendered image and the reconstructed 3D geometry.

## 4.1 Multi-view 3D reconstruction

**DTU** The DTU [12] dataset contains multi-view image (49 or 64) of different objects with fixed camera and lighting parameters. We evaluate our method on the 15 scans that were selected by [38]. We compare our surface accuracy using the Chamfer $l_1$ loss (measured in mm) to COLMAP$_0$ (which is watertight reconstruction; COLMAP$_7$ is not watertight and provided only for reference) [31], NeRF [21] and IDR [38], where for fair comparison with IDR we only evaluate the reconstruction inside the visual hull of the objects (defined by the segmentation masks of [38]). We further evaluate the PSNR of our rendering compared to [21]. Quantitative results are presented in Table 1. It can be observed that our method is on par with IDR (that uses object masks for all images) and outperforms NeRF and COLMAP in terms of reconstruction accuracy. Our rendering quality is comparable to NeRF's.

**BlendedMVS** The BlendedMVS dataset [37] contains a large collection of 113 scenes captured from multiple views. It supplies high quality ground truth 3D models for evaluation, various camera configurations, and a variety of indoor/outdoor real environments. We selected 9 different scenes and used our method to reconstruct the surface of each object. In contrast to the DTU dataset, BlendedMVS scenes have complex backgrounds. Therefore we use NeRF++ [39] as a baseline for this dataset. In Table 2 we present our results compared to NeRF++. Qualitative comparisons are presented in Fig. 5; since the units are unknown in this case we present relative improvement of Chamfer distance (in %) compared to NeRF. Also in this case, we improve NeRF reconstructions considerably, while being on-par in terms of the rendering quality (PSNR).

**Comparison to [38]** IDR [38] is the state of the art 3D surface reconstruction method using implicit representation. However, it suffers from two drawbacks: first, it requires object masks for training, which is a strong supervision signal. Second, since it sets the pixel color based only on the single point of intersection of the corresponding viewing ray, it is more pruned to local minima that sometimes appear in the form of extraneous surface parts. Figure 6 compares the same scene trained

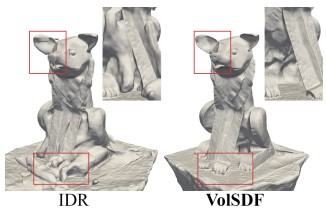

Figure 6: IDR extraneous parts.

| | Scene | Doll | Egg | Head | Angel | Bull | Robot | Dog | Bread | Camera | **Mean** |
|---|---|---|---|---|---|---|---|---|---|---|---|
| Chamfer $l_1$ | Our Improvement (%) | 54.0 | 91.2 | 24.3 | 75.1 | 60.7 | 27.2 | 47.7 | 34.6 | 51.8 | 51.8 |
| PSNR | **NeRF++** | 26.95 | 27.34 | 27.23 | 30.06 | 26.65 | 26.73 | 27.90 | 31.68 | 23.44 | 27.55 |
| | **VolSDF** | 25.49 | 27.18 | 26.36 | 29.79 | 26.01 | 26.03 | 28.65 | 31.24 | 22.97 | 27.08 |

Table 2: Quantitative results for the BlendedMVS dataset.

with IDR with the addition of ground truth masks, and VolSDF trained without masks. Note that IDR introduces some extraneous surface parts (e.g., in marked red), while VolSDF provides a more faithful result in this case.

## 4.2 Disentanglement of geometry and appearance

We have tested the disentanglement of scenes to geometry (density) and appearance (radiance field) by switching the radiance fields of two trained scenes. For VolSDF we switched $L_\psi$. For NeRF [21] we note that the radiance field is computed as $L_\psi(z, v)$, where $L_\psi$ is a fully connected network with one hidden layer (of width 128 and ReLU activation) and $z$ is a feature vector. We tested two versions of NeRF disentanglement: First, by switching the original radiance fields $L_\psi$ of trained NeRF networks. Second, by switching the radiance fields of trained NeRF models with an identical radiance field model to ours, namely $L_\psi(x, n, v, z)$. As shown in Figure 7 both versions of NeRF fail to produce a correct disentanglement in these scenes, while VolSDF successfully switches the materials of the two objects. We attribute this to the specific inductive bias injected with the use of the density in equation 2.

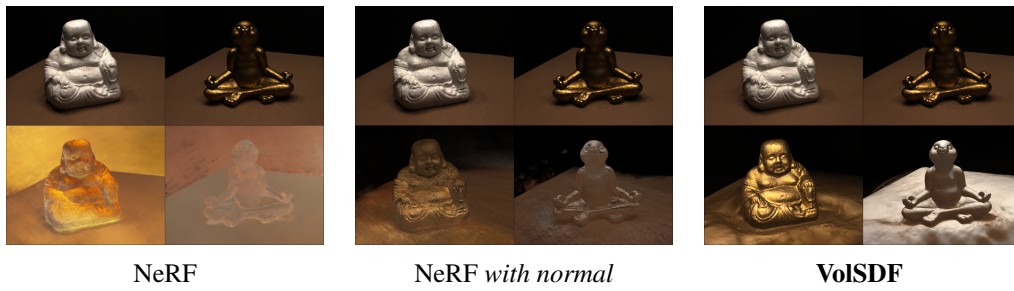

| NeRF | NeRF *with normal* | **VolSDF** |

Figure 7: Geometry and radiance disentanglement is physically plausible with VolSDF.

## 5 Conclusions

We introduce VolSDF, a volume rendering framework for implicit neural surfaces. We represent the volume density as a transformed version of the signed distance function to the learned surface geometry. This seemingly simple definition provides a useful inductive bias, allowing disentanglement of geometry (i.e., density) and radiance field, and improves the geometry approximation over previous neural volume rendering techniques. Furthermore, it allows to bound the opacity approximation error leading to high fidelity sampling of the volume rendering integral.

Some limitations of our method present interesting future research opportunities. First, although working well in practice, we do not have a proof of correctness for the sampling algorithm. We believe providing such a proof, or finding a version of this algorithm that has a proof would be a useful contribution. In general, we believe working with bounds in volume rendering could improve learning and disentanglement and push the field forward. Second, representing non-watertight manifolds and/or manifolds with boundaries, such as zero thickness surfaces, is not possible with an SDF. Generalizations such as multiple implicits and unsigned fields could be proven valuable. Third, our current formulation assumes homogeneous density; extending it to more general density models would allow representing a broader class of geometries. Fourth, now that high quality geometries can be learned in an unsupervised manner it will be interesting to learn dynamic geometries and shape spaces directly from collections of images. Lastly, although we don't see immediate negative societal impact of our work, we do note that accurate geometry reconstruction from images can be used for malice purposes.

## Acknowledgments

LY is supported by the European Research Council (ERC Consolidator Grant, "LiftMatch" 771136), the Israel Science Foundation (Grant No. 1830/17), and Carolito Stiftung (WAIC). YK is supported by the U.S.- Israel Binational Science Foundation, grant number 2018680, Carolito Stiftung (WAIC), and by the Kahn foundation.

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
