# OpenReview forum: "Volume Rendering of Neural Implicit Surfaces"
_NeurIPS.cc/2021/Conference — NeurIPS 2021 Oral_

### Official Review · Reviewer_ybo7 · 2021-07-04

**Rating:** 9
**Confidence:** 4

**Summary:**

The paper introduces a novel method for multi-view scene reconstruction and image synthesis. The authors combine volumetric and surface based neural rendering techniques to obtain highly accurate surface models without need for object mask annotation. The paper presents theoretical framework motivating the SDF-based density field parametrization and the adaptive sampling approach. The validation shows that the reconstructed shapes are surpassing the state of the art.

**Ethical Concerns:**

Do not see any.


**Limitations And Societal Impact:**

[Societal impact]
The authors (and me) do not expect negative societal impacts. It could potentially lead to more effective 3D data capture, storage and transfer which is useful in many fields including cultural heritage preservation.



[Limitations]
The paper lacks listing of limitations (Or I could not find it). Perhaps with the exception of future work discussion in the conclusion. In the video I noticed some instability of the ray-traced surface in the skull example (may or may not be related to the representation) and potentially lack of spatial fidelity of the reconstructed image details. The method also shares the encoding/decoding computational complexity with its predecessors.


**Main Review:**

[Originality]
The idea of parametrizing volumetric density through signed distance function is novel, very elegant and effective. I expect it to have significant impact on the community.

[Quality]
The algorithms are well motivated and based on theoretical analysis.

[Clarity]
The paper is clearly written and easy to follow. I particularly like the elegant definition of the SDF and surface manifold in Eq. 1. The proofs of theoretical relations from the paper are provided in the supplement though they could be expanded to cover more details. E.g. I am not entirely sure how the second inequality in the proof of Lemma 2 emerges from the derivation of Lemma 1.
The paper covers implementation details needed for reproduction and, furthermore, code was also provided.

[Significance]
The capability of surface-based inverse neural rendering without mask requirement lifts one of the major practical limitations from this category of methods and it should allow for easier and higher quality surface reconstruction methods in the future.

[Questions, Comments and Concerns]

1) Why is the density around the surface modeled using Laplacian distribution? Is there a physical reason for this particular distribution or is it just mathematical convenience?

2) The normal as input parameter of L makes sense but I would still expect explicit ablation testing the same model with and without this new feature. Does it affect reconstruction quality or is it only useful for the appearance swap demo in Sec 4.2?

3) The quantitative evaluation does not fully capture the image reconstruction quality. The shape reconstruction without masks alone is a great achievement but it would still be useful to learn limits of the image reconstruction quality. The PSNR scores seem competitive with Nerf(++), however, closer inspection of Fig. 5 shows that image features are rather smeared. The images are also of relatively low resolution. It would be useful to provide a metric more sensitive to perceptual qualities of the images such as PSNR. Further, how would the proposed technique compare to some of the recent surface-based techniques focusing on image reconstruction quality such as "Stable View Synthesis" [Riegler and Koltun 2021]? (I believe it should also be cited)


4) Minor comments:
- L122: transperancy -> transparency
- L209: while keep -> while keeping


[Conclusion]

Overall, this is a very high quality and high impact potential paper. I believe that the proposed hybrid formulation is a very promising way forward for surface reconstruction methods. My only notable concern with the paper is the extent of image reconstruction quality evaluation. However, it does not have a direct impact on the main contribution and it can easily be improved in the final version. I strongly recommend accepting.


**Time Spent Reviewing:**

7

---

> ### Author Response · Authors · 2021-08-10
> **Authors response**
>
> We thank the reviewer for his insightful and thorough review. We will further incorporate other suggestions of the reviewer in the next version. In the following we address the main concerns raised in this review.
>
> We provide extra results to answer questions raised by the reviewers in the link - https://sites.google.com/view/volsdf.
>
> ***Q: “Why is the density around the surface modeled using Laplacian distribution? Is there a physical reason for this particular distribution or is it just mathematical convenience?”*** \
> A:
> Theoretically, any CDF/PDF pair with a scale parameter (e.g., normal distribution CDF/PDF) would transform an SDF to occupancy and will furnish some corresponding bound. The benefit in the Laplace PDF is in its behaviour near zero, where it decreases with a $\beta^{-2}$ slope.
> Physically, the Laplace CDF provides a connection between distance functions and certain solutions of the screen poisson equation.  We will add a discussion on different possible distributions and their bounds.
>
>
>
> ***Q: “The normal as input parameter of L makes sense but I would still expect explicit ablation testing the same model with and without this new feature. Does it affect reconstruction quality or is it only useful for the appearance swap demo in Sec 4.2?”*** \
> A:
> Incorporation of the normal in $L$ improves both reconstruction quality and appearance swap. To demonstrate this directly we provide the relevant ablation in Figure B6 in https://sites.google.com/view/volsdf . We will add this ablation to the supplementary of the paper.
>
> ***Q: “The quantitative evaluation does not fully capture the image reconstruction quality.”*** \
> A:
> We unfortunately do not fully understand what evaluation metric the reviewer suggests: could the second “PSNR” mentioned in this question be a typo?  In case the reviewer means “LPIPS” - we evaluated LPIPS of the image reconstruction of the DTU dataset and got mean result of 0.38 for NeRF and 0.39 for VolSDF. If that is indeed the reviewer's intention we will add the per-scene LPIPS evaluation to the supplementary.
>
> ***Q: “Closer inception of figure 5 shows that Image features are rather smeared”*** \
> A:
> One explanation to this phenomenon can be the usage of level 6 positional encoding in VolSDF compared to level 10 used in NeRF. In Figure B3 in the link https://sites.google.com/view/volsdf we show the DTU Bunny scene with positional encoding levels 6 and 10 for both NeRF and VolSDF; we report both PSNR for the rendered images and Chamfer distance to the learned surfaces. Note that higher positional encoding improves specular highlights and details of VolSDF but adds some undesired noise to the reconstructed surface.
>
> ***Q: “The paper lacks listing of limitations (Or I could not find it). Perhaps with the exception of future work discussion in the conclusion.”*** \
> A:
> In the supplementary, figure A2 and the text next to it (Lines 18-33) depicts and explains the main failure cases of our method
>
> ***Q: ”In the video I noticed some instability of the ray-traced surface in the skull example (may or may not be related to the representation) and potentially lack of spatial fidelity of the reconstructed image details.”*** \
> A:
> We thank the reviewer for this comment. Instability of the ray-traced surfaces is due to a bug we had in finding the intersection point while creating these videos. A corrected version is demonstrated in Figure B7 in https://sites.google.com/view/volsdf.
>
> ***Q: ”The method also shares the encoding/decoding computational complexity with its predecessors.”*** \
> A:
> We agree and we clarify this in the limitations.

---

> > ### Comment · Reviewer_ybo7 · 2021-08-13
> > **Rebuttal Feedback**
> >
> > Thank you. This addresses my concerns.

---

### Official Review · Reviewer_EPLB · 2021-07-13

**Rating:** 7
**Confidence:** 5

**Summary:**

This paper presents a neural approach based on NeRF, for the purpose of learning a 3D representation of a shape from captured images. Instead of allowing volume density to be freely allocated along rays, as in NeRF, this paper proposes concentrating volume density around a single surface. To do this, the method represents the shape as a neural SDF, and defines a volumetric density around the zero level set of the SDF using the CDF of the Laplace distribution. This choice enables the paper to derive a bound on the estimated opacity for any segment along a ray, and this is used to draw samples such that the error is limited to some epsilon. The paper experimentally shows that the resulting recovered surfaces are closer to the ground-truth geometry than surfaces extracted from level sets of NeRF's volumetric geometry, as well as meshes produced by COLMAP. Additionally, the paper experimentally shows that the proposed method is able to render novel views that are close in quality to those rendered by NeRF.

**Limitations And Societal Impact:**

The paper includes an interesting discussion of potential future directions and a figure with example failure cases. I think that it would be nice to include a more thorough discussion of limitations in the updated draft.

**Main Review:**

I like the paper and the general research direction, but I have some concerns regarding the experimental evaluation.

Strengths:

Overall, think that the idea of hybrid surface-volume representation and rendering is well-motivated, and this paper develops a well thought-out approach to recover NeRF-like models whose volume densities are concentrated around a surface. I intellectually like the idea of choosing a volumetric function of the SDF that lets us bound the opacity estimation.

The proposed method clearly outperforms IDR and level sets extracted from NeRF, in terms of surface reconstruction quality.

I think that the disentanglement results are very cool and thought-provoking.

Weaknesses:

I think that the main weaknesses of the paper are in the thoroughness of the experimental evaluation. I am listing a few of my questions/concerns below:

It is unclear how many actual MLP evaluations along a ray are used for each of the baselines and ablations in A1. I am concerned that the experiments are not controlling for the number of MLP evaluations along each ray when comparing different methods. For example, is it correct that VolSDF with 5 iterations is using more than 832 MLP evaluations per ray (5*128 in order to compute the error bounds for each sample for each iteration + 128 for estimating \hat{O} + final 64 "fresh" samples, and this is without accounting for the "upsampling" done in each iteration)? If this is true, then I think that the comparisons are potentially quite unfair, because NeRF results would probably be much better with this number of samples per ray (instead of 64+128 as in the default NeRF implementation). Could the authors please clarify this?

I think it would be useful to make the ablation figure in A1 into a more complete comparison across multiple scenes, and additionally clarify how many actual MLP queries along a ray are used by each of these methods, so readers can form a reasonable basis for comparison.

I've noticed these same noisy salt-and-pepper artifacts (like those seen in Figure 2 and mentioned on line 158) when experimenting with NeRF if I use stratified random sampling of points along the ray for the "coarse" MLP at test time (as opposed to training time, where NeRF draws random stratified samples along the ray for the "coarse" MLP, at test time, NeRF uses regularly-spaced samples for the "coarse" MLP). Is it possible that these artifacts are actually due to the NeRF implementation forgetting to turn off random sampling at test time?

The scenes shown in the supplementary video are composed of materials that have pretty simple view-dependent appearance (not too shiny and not illuminated by obviously high-frequency lighting). Furthermore, they are only rendered from a limited range of views, so the view-dependent appearance showcased in these results is not very complex. I would be very interested to see how this method performs on scenes like the Blender material ball samples (in NeRF's Blender dataset) that are much shinier and contain reflections of high-frequency lighting. Also, I'd suggest including rendered camera path results from the BlendedMVS dataset and camera paths with much larger motion (full 360s around an object maybe?) in the supplementary video, because it would help readers get a better idea how VolSDF performs.

I'm confused by the statement on line 290, which says that NeRF computes L(z,v) with a linear function. My understanding is that NeRF uses a single layer with a ReLU linearity (width = 128) that takes in z and v, followed by a linear layer that outputs the view-dependent emitted radiance RGB value. Could the authors please comment on the NeRF baseline implementation?

Another concern I have regarding comparisons to the NeRF baseline: line 59 of the supplementary PDF says that the VolSDF model has 4 layers of width 256 that represent the view-dependent emitted radiance. I think it would make sense to use the same higher-capacity architecture for the NeRF baseline for the experiments in Table 1 and the disentanglement comparisons.

A few other questions:

Does the representation still need to be initialized as a unit sphere SDF (supplementary line 57) in this hybrid surface/volume rendering setup? I would guess that this initialization is less important than it would be in a purely surface-based representation like IDR.

My understanding is that the alpha and beta parameters of the Laplace CDF are global for the whole scene. Would it make sense to instead have them be local, so that different regions of the scene could be more volumetric vs. more surface-like?

The choice of setting alpha to the reciprocal of beta seems unintuitive, can the authors please elaborate why both can't be free parameters during optimization? What happens to the results if both are free?

Edits and suggestions:

Since the main goal of this paper is to improve the surface representation recovered from NeRF-like models (as opposed to improving view synthesis quality), I think it would be good to include more motivation why we should care about better surface reconstruction from these methods. Maybe mentioning robotics applications or other scenarios where we would like to use differentiable rendering to recover geometry for some application that actually needs high-quality geometry (one could argue that in view synthesis, geometry doesn't matter if the views are rendered well)?

I think that the usage of the term "light field" in this paper is slightly different from how it is typically used in graphics, and this may be confusing to readers. This paper uses "light field" to mean the view-dependent emitted radiance function in NeRF (how much radiance is emitted in any direction at each point), which is a quantity that is accumulated to render the radiance travelling along a ray. However, the typical usage of "light field" within graphics (see [Levoy and Hanrahan, SIGGRAPH 1996]) refers to the radiance along rays, which is actually the quantity that is produced from volume rendering. These usages of the term "light field" are slightly different, because the actual light field function would represent the accumulated opacity along all rays instead of just what is being emitted by particles at each location. I suggest changing the usage of "light field" in this paper to avoid confusion.

Line 22-23: remove "so called"

Line 69: add dash between non and Lambertian

Lines 105, 139, 147, etc.: capitalize "Equation" to match capitalization of "Figure" and "Section"

Line 114: it --> the volume rendering integral

First line of Fig. 3 caption: missing space after "1"

Line 313: malice --> malicious

I'd recommend proofreading the references. For example, CVPR is cited differently in [12] and [14], [17] was published at SIGGRAPH, NeRF [21] should have the R and F capitalized, NeRD [2] should have the R and D capitalized, NeRV [34] should have the R and V capitalized, the Adam optimizer was published at ICLR and not just arXiv, etc.

**Time Spent Reviewing:**

4

---

> ### Author Response · Authors · 2021-08-10
> **Authors response**
>
> We thank the reviewer for his insightful and thorough review. We will further incorporate other suggestions of the reviewer in the next version. In the following we address the main concerns raised in this review.
>
> We provide extra results to answer questions raised by the reviewers in the link - https://sites.google.com/view/volsdf.
>
> ***Q: “I am concerned that the experiments are not controlling for the number of MLP evaluations along each ray when comparing different methods.”*** \
> A:
> Our algorithm uses 128*5=640 MLP forward evaluations and additional 64 forward+backward (MLP evaluations and gradient computations) samples per ray. NeRF uses (64+192=)256 forward+backward samples. This provides a comparable iteration time (when both use gradient information), which was the basis to our evaluation. To further address the reviewer’s concern we also ran an additional experiment where we used NeRF sampling (coarse and fine) where we used 640 forward samples of the coarse network and 64 forward+backward samples, similar to our MLP evaluation budget. Figure B4 in https://sites.google.com/view/volsdf shows qualitative as well as quantitative (i.e., PSNR, Chamfer and time evaluation) results.
>
>
> ***Q: “...it would be useful to make the ablation figure in A1 into a more complete comparison across multiple scenes”.*** \
> A:
> We will add more scenes to the ablation test.
>
> ***Q: “Is it possible that these artifacts are actually due to the NeRF implementation forgetting to turn off random sampling at test time?”*** \
> A:
> Both random sampling and regularly-spaced sampling in NeRF introduce artifacts. While it’s true that salt-and-pepper artifacts are due to the stochastic nature of the sampling, using regularly-spaced sampling introduces a different kind of errors and artifacts, as can be seen in Figure B5 in https://sites.google.com/view/volsdf.
>
> ***Q: “I would be very interested to see how this method performs on scenes like the Blender material ball samples that are much shinier and contain reflections of high-frequency lighting. Also, I'd suggest including rendered camera path results from the BlendedMVS dataset and camera paths with much larger motion (full 360s around an object maybe?)”*** \
> A:
> Figures B2(bottom) and B1 in https://sites.google.com/view/volsdf present the results of the requested experiments, namely the Blended material ball and 360 degrees camera path for one scene from Blended MVS dataset. We will add those to the supplementary video.
> Lastly, note that the reason that the supplementary video is rendered from a limited range of views is because the input views in the DTU dataset is limited, the unseen back of the object is given as an example in figure A2 in the supplementary.
>
> ***Q: Is $L(z,v)$ in nerf a linear function?*** \
> A:
> Thank you for finding this error in the text (the code for NeRF was correct), we will fix this in the paper.
>
> ***Q: “I think it would make sense to use the same higher-capacity architecture for the NeRF baseline for the experiments in Table 1 and the disentanglement comparisons.”*** \
> A:
> In the disentanglement experiment we used the *same* architecture for VolSDF and NeRF, see Lines 292-293. In Table 1 we used the official NeRF implementation. Note however, that NeRF produces rendered images with PSNR comparable to VolSDF, so the capacity of the NeRF model does not seem to be the bottleneck in its inferior geometry reconstruction.
>
> ***Q: Spatially varying alpha,beta?*** \
> A:
> We consider this as an interesting future work. We will add it to the future work directions.
>
> ***Q: Why $\alpha=\beta^{-1}$? What happens when both are free?*** \
> A:
> The choice $\alpha=\beta^{-1}$ models opaque homogeneous material objects. When $\alpha,\beta$ are free, VolSDF could potentially represent more complicated materials; experimentally, in all the datasets that we considered, we didn’t find sufficient justification for that more general model and hence did not further explore this option.
>
> ***Q: “Edits and suggestions.”*** \
> A:
> We will use reviewer suggestions to further improve the next version of the paper.

---

> > ### Comment · Reviewer_EPLB · 2021-08-30
> > **Response**
> >
> > Thank you for the detailed response to my questions and concerns.
> >
> > I appreciate the clarification that the sample counts for comparison are based on controlling for iteration time, as well as the additional experimental results for NeRF with this increased sample count. I'm also curious how VolSDF would perform with just 256 samples total.
> >
> > Regarding the results of VolSDF on NeRF's Blender scenes, it seems like VolSDF produces much blurrier/low-frequency renderings than NeRF, particularly on the chrome ball reflections and the treads of the lego truck. Could the authors please comment?
> >
> > Overall, I still like this paper and am happy to recommend that it be accepted.

---

> > > ### Author Response · Authors · 2021-09-01
> > > **Response**
> > >
> > > Thank you for your reply and extra questions.
> > >
> > > ***Q: "I'm also curious how VolSDF would perform with just 256 samples total.”*** \
> > > A: Two iterations of VolSDF uses 256 samples in total and produce chamfer distance 0.73 for the skull scene (used in the ablation experiment). The full VolSDF used 640 samples and produced a chamfer distance of 0.7. In comparison NeRF using 640 samples in total produced a chamfer distance of 1.12. While the 256 and 640 samples versions of VolSDF produced similar chamfer distance, visually the 640 version is slightly better.
> > >
> > > ***Q: "Regarding the results of VolSDF on NeRF's Blender scenes, it seems like VolSDF produces much blurrier/low-frequency renderings than NeRF, particularly on the chrome ball reflections and the treads of the lego truck. Could the authors please comment?"*** \
> > > A: We believe the VolSDF-NeRF tradeoff is as follows. VolSDF produces smoother, higher detail, lower noise surfaces, while level sets of NeRF’s density tend to be very noisy in general. On the other hand, this noise (with or without the help of higher PE frequency) could potentially explain the rendering of sharper images with NeRF; see for example the geometry of the chrome balls in the NeRF results in Figure B2 in https://sites.google.com/view/volsdf corresponding to different reflections, where balls with more detailed reflections correspond to higher noise in the geometry.

---

### Official Review · Reviewer_L3KT · 2021-07-16

**Rating:** 9
**Confidence:** 3

**Summary:**

NeRF and follow-up papers have led to impressive improvements in view synthesis when trained from multiple views of a single scene. This is achieved through volumetric rendering of an opacity field and a decoupled reflectance network. A limitation of NeRF is that the opacity field might not be faithful to the true geometry, as long as the rendered RGB images reproduce the correct appearance.

The proposed work builds on ideas developed in the NeRF literature and adapts them to 3D surface reconstruction. Contributions include deriving opacity from a truncated signed distance function, which explicity models the inside-outside topology of the surface. The validity of the signed distance function is enforced though an Eikonal loss term. The authors present a novel sampling technique for volumetric rendering, based on an approximation of the current opacity with bounded error.


**Ethical Concerns:**

The authors have adequately adressed the ethical concerns surrounding this kind of work.


**Limitations And Societal Impact:**

The potential for negative societal impact of this work is adequately discussed. See Main Review for a discussion of other limitations.

**Main Review:**

I consider this paper a strong submission that clearly identifies the limited applicability of NeRF-like goemetry representations to surface reconstruction. The authors present an intuitive geometry representation based on a truncated signed distance function, and an opacity field that is derived from the distance using a Laplace distribution. The width of the distribution is a learnable global parameter, which appears to allow for a gradual optimization from a relatively transparent initial geometry, to a well-defined, fully-opaque surface upon convergence.

The paper includes a thorough theoretical discussion of an adaptive sampling technique along the viewing ray, as well as a proof of error bounds on the opacity estimate that is used for sampling. The sampling technique is verified empirically in the supplemental material.

The presented experiments are comprehensive, comparing the accuracy of the reconstructed geometry to a state-of-the-art traditional technique (COLMAP), a recent learned technique (IDR), as well as a surface extracted from NeRF's opacity field. Both qualitatively and quantitatively, the presented 3D reconstructions compare very favourably to COLMAP and NeRF. It is exciting that the technique does not require silhouette masks as input, which promises to significantly increase the technique's practical viability.

The paper also includes an experiment that demonstrates improved disentanglement of geometry and reflectance compared to NeRF. It is surprising to this reviewer that the radiance network yields reliable predictions when sampled for positions near the surface of the new geometry, even though at training time the samples are placed near the original surface (guided by adaptive sampling). Could the authors comment why this apparent lack of supervision is not an issue?

### Limitations

I would expect a surface-based representation to show limitations for reconstruction problems that involve scenes without clearly defined surfaces, for example smoke or flames. The presented results do not include results on highly detailed geometry, such as hair or plants, which would require a level of detail that might exceed the resolution of the SDF, but could be represented with "soft visibility" for parts of the scene. Could the authors elaborate on the design space between representations with unconstrained opacity, and surface-based representation?

In direct comparison to baseline NeRF (see video), novel views appear to have somewhat smoother geometry and duller specular highlights. The paper would benefit from an analysis of implicit smoothness priors in the current implementation (Eikonal loss? L1 regularization in opacity transform?)

### Further comments

 - The relative impact of the Eikonal loss term should be verified with an ablation study.

 - The sampling ablation from the supplemental would be useful addition to the main paper.




**Time Spent Reviewing:**

6

---

> ### Author Response · Authors · 2021-08-10
> **Authors response**
>
> We thank the reviewer for his insightful and thorough review. We will further incorporate other suggestions of the reviewer in the next version. In the following we address the main concerns raised in this review.
>
> We provide extra results to answer questions raised by the reviewers in the link - https://sites.google.com/view/volsdf.
>
> ***Q: “It is surprising to this reviewer that the radiance network yields reliable predictions when sampled for positions near the surface of the new geometry, even though at training time the samples are placed near the original surface (guided by adaptive sampling). Could the authors comment why this apparent lack of supervision is not an issue?”*** \
> A:
> This is a good question. We attribute this phenomena to two facts: First, all the DTU objects have *roughly* the same location. Second, the BRDF of homogeneous materials can be represented as a simple (i.e., nearly constant) function of the normal with no dependence on the location. Providing the light field $L$ with the normal information (see Equation 7) allows the model to learn this simple dependency that generalizes beyond the original surface’s geometry.
>
>
> ***Q: “The presented results do not include results on highly detailed geometry, such as hair or plants”; “Could the authors elaborate on the design space between representations with unconstrained opacity, and surface-based representation?”*** \
> A:
> This is a great question and worthy of follow-up work; we will add this to the conclusion section of the paper. In essence, representing non-watertight manifolds and/or manifolds with boundaries, such as zero thickness surfaces, is not possible with an SDF. Generalization to the method can include one or more of the following: multiple implicits, unsigned fields, and spatially varying density and width parameters ($\alpha,\beta$). These can generalize the homogeneous material case that is explored in our paper and modeled with a single SDF. The challenge would be to keep the sampling guarantees. We also agree it would be interesting to understand the limit of such representations and in turn augment them to increase the capacity to include more geometry/density types.
> Regarding examples with high details we note that many of the scenes in the paper contain fine details, e.g., the Gundam and Doll models in Figure 5. We further ran VolSDF on the synthetic datasets of NeRF and the results can be found in Figure B2 in the following link: https://sites.google.com/view/volsdf .
>
> ***Q: “In direct comparison to baseline NeRF...novel views appear to have somewhat smoother geometry and duller specular highlights”*** \
> A:
> One explanation to this phenomenon can be the usage of level 6 positional encoding in VolSDF compared to level 10 used in NeRF. In Figure B3 in the link https://sites.google.com/view/volsdf we show the DTU Bunny scene with positional encoding levels 6 and 10 for both NeRF and VolSDF; we report both PSNR for the rendered images and Chamfer distance to the learned surfaces. Note that higher positional encoding improves specular highlights and details of VolSDF but adds some undesired noise to the reconstructed surface.
>
>
> ***Q: Sampling ablation should be in the main paper.*** \
> A:
> We will move the ablation into the main paper in the next version.

---

> > ### Comment · Reviewer_L3KT · 2021-08-18
> > **Reviewer feedback**
> >
> > Figure B3 in (comparison between level 6 and level 10 encoding) is insightful. I think it would be important for the paper to point out that the quantitative chamfer results are sensitive to the frequency of the positional encoding (and possibly other smoothness constraints). This is not really surprising, given that object geometry is not directly supervised during training, but it seems like an important aspect of the implementation.
> >
> > Readers who are interested in view synthesis will be pleased to see that VolSDF appears to be able to match the synthesis quality (both in terms of PSNR and perceptually) of baseline NeRF when L10 positional encoding is used. While it's a bit unfortunate that there is not one set of hyper parameters that will minimize all metrics, I believe that the quantitative results will be better understood by readers if this trade-off is made more explicit.
> >
> > Besides this, all my questions have been sufficiently addressed by the authors' response.

---

> > > ### Author Response · Authors · 2021-08-19
> > > **Thanks for your feedback!**
> > >
> > > Thank you for your suggestions and help in improving the paper. We will incorporate this comparison and trade-off discussion in the revised paper.

---

### Official Review · Reviewer_rpsv · 2021-07-19

**Rating:** 9
**Confidence:** 4

**Summary:**

The paper proposed a method to recover both geometry and appearance of objects. They
achieve this by combing both SDF representations and volume rendering. Given a set of
multi-view images, they use a network to predict the SDF and radiance filed of an arbitrary point, which
is mapped to the volume density using a bell-shaped analytic function. Then they perform
volume rendering to predict the color for the pixel. They minimize the color difference as
well as an Eikonal loss to train the networks. They also propose a sampling scheme to  adaptively sample points along the ray based on the derived theoretical bound. The results
show that the proposed method can generate both accurate 3D mesh and faithful view-dependent
appearance.


**Limitations And Societal Impact:**

Please see the main review.

**Main Review:**


### Strengths
1. Compared to previous volume-based methods, the proposed method can generate better mesh for the
captures scenes and produces comparable appearance.

2. Compared to previous SDF-based methods, the proposed method can accurate reconstruct the geometry
with the need for foreground masks, which makes the acquisition setup much simpler.

3. The idea of mapping SDF to volume density is interesting and novel.

4. The paper develops a novel method for adaptively sampling points along the ray, and it gives
a theoretical proof for the sampling scheme. Such a sampling scheme get rids of using coarse-fine
networks as in NeRF, and also leads to better accuracy.

### Weakness
1. In the sampling scheme, it's not clear what the upsampling means exactly. It seems to me
that the method is gradually adding new samples to the current sampling points. My concern is
that whether such a scheme will cause memory and efficiency problem when there are many sampling
points considering that the network needs to be evaluated at each point. How much time does
the sampling take?

2. As far as I understanding, after the sampling iterations finish, the sampled points
$\mathcal{T}$ and $\beta_{+}$ satisfying the desired bound. However, in Line 11, the algorithm
generates a new set of samples. Do this new set of samples also meet the bound? If not,
how does this affect the performance? Also when there are more than 64 intervals, how these
$64$ samples are generated? Does it mean some of the intervals will have zero samples?

3. How does the method work for scenes with a lot of self-occlusions and thin structures?
Most of the objects in the scene have relatively simple convex geometry.  It is desirable to show
results on the NeRF dataset such as the Lego, the Ship and the Plant scene.

4. The paper only shows object-level reconstructions. Since the method does not need image masks,
I would expect the method to also work on complex scenes.  How does the method work on large-scale scenes as those shows in the NeRF paper?

5. In the results in Figure A4 (supp), it seems that the recovered appearance tends to lose
details and is a bit blurry, as shown in the Bull and the Dog scene. What is the reason
for it?

6. The proposed method can generate accurate meshes from the SDF. One question is that whether it's possible to direct use the colors of points on the isosurface to represent the appearance of the objects? Or volume rendering is still needed to predict the colors?

7. Another limitation of the highlighted is that the texture/normal ambiguity. The method tends
to treat the texture changes as normal changes in the reconstructions, such as
the floor in the second scene of Figure 5.

Overall, I appreciate the idea of the paper, and the results shown in the paper are convincing.
I expect the authors to answer the questions above and add further clarifications to the paper.



**Time Spent Reviewing:**

3.5

---

> ### Author Response · Authors · 2021-08-10
> **Authors response**
>
> We thank the reviewer for his insightful and thorough review. We will further incorporate other suggestions of the reviewer in the next version. In the following we address the main concerns raised in this review.
>
> We provide extra results to answer questions raised by the reviewers in the link - https://sites.google.com/view/volsdf.
>
> ***Q: “Not clear what upsampling means...How much time the sampling takes?”*** \
> A:
> The upsampling procedure is defined in lines 209-212; we will further clarify the upsampling procedure in the next version. In essence: our sampling algorithm performs 128*5=640 MLP forward passes in total. The sampling takes approximately 0.15 seconds, where the iteration time takes approximately 0.22 seconds (the number computed on 3 different DTU scenes experiments).
>
> ***Q: Does the new sample set generated in line 11 of the algorithm satisfy the bound?*** \
> A:
> The set $\mathcal{T}$ satisfies the bound and therefore the set $\mathcal{S}$ is a good sample set for approximating the volume rendering integral in Equation 7 via Equation 8. An interesting future work will be to develop an error bound for equation 8, however that will require developing a method to bound integration error of the radiance field.
>
>
> ***Q: “How does the method work for scenes with a lot of self-occlusions and thin structures?”*** \
> A:
> Many of the scenes in the paper are not convex and contain thin structures; for example, the Gundam’s wings (second from the left, and in the following link) and the Doll’s hat (right) in Figure 5 and the scissors in Figure A2(c) in the appendix. We further ran VolSDF on the synthetic datasets of NeRF and the results can be found in Figure B2 in the following link: https://sites.google.com/view/volsdf .
>
> ***Q: “The paper only shows object-level reconstructions. Since the method does not need image masks, I would expect the method to also work on complex scenes. How does the method work on large-scale scenes as those shows in the NeRF paper?”*** \
> A:
> Most of the complex scenes are forward facing scenes which are represented in NeRF using NDC (Normalized device coordinate) space; adapting implicit neural representations to this case is marked as an interesting future work.
>
> ***Q: “In the results in Figure A4 (supp), it seems that the recovered appearance tends to lose details and is a bit blurry, as shown in the Bull and the Dog scene. What is the reason for it?”*** \
> A:
> One explanation to this phenomenon can be the usage of level 6 positional encoding in VolSDF compared to level 10 used in NeRF. In Figure B3 in the link https://sites.google.com/view/volsdf we show the DTU Bunny scene with positional encoding levels 6 and 10 for both NeRF and VolSDF; we report both PSNR for the rendered images and Chamfer distance to the learned surfaces. Note that higher positional encoding improves specular highlights and details of VolSDF but adds some undesired noise to the reconstructed surface.
>
> ***Q: “Is it possible to directly use the colors of points on the isosurface to represent the appearance of the objects?”*** \
> A:
> It is possible to use surface rendering, however it requires finding the surface-ray intersection precisely. Note that as beta approaches zero the volume rendering will converge to surface rendering where all density $\tau$ will be concentrated at the intersection point.
>
>
> ***Q: Limitation of confusing texture with normal.*** \
> A:
> Thank you for this comment, we will add this to the limitations of our method.

---

> > ### Comment · Reviewer_rpsv · 2021-08-20
> > **Reviewer feedback**
> >
> > Thank you. The rebuttal addresses my concerns.

---

### Decision · Program_Chairs · 2021-09-27

**Decision:**

Accept (Oral)

**Comment:**

This submission introduces a timely and important contribution that integrates SDF representations with neural volume rendering.  All reviewers are positive, and three of the four reviewers strongly recommend acceptance.  The AC agrees.  The authors should try to address the reviewers' concerns in the camera-ready version. This includes adding the ablation studies from the rebuttal period, incorporating the discussion as L3KT suggested, among others.